# Macrophage Migration Inhibitory Factor (MIF) Plasma Concentration in Critically Ill COVID-19 Patients: A Prospective Observational Study

**DOI:** 10.3390/diagnostics11020332

**Published:** 2021-02-17

**Authors:** Christian Bleilevens, Josefin Soppert, Adrian Hoffmann, Thomas Breuer, Jürgen Bernhagen, Lukas Martin, Lara Stiehler, Gernot Marx, Michael Dreher, Christian Stoppe, Tim-Philipp Simon

**Affiliations:** 1Department of Anesthesiology, Medical Faculty Rheinisch-Westfälische Technische Hochschule Aachen University Hospital, D-52074 Aachen, Germany; 2Department of Intensive Care and Intermediate Care, Medical Faculty Rheinisch-Westfälische Technische Hochschule Aachen University Hospital, D-52074 Aachen, Germany; jsoppert@ukaachen.de (J.S.); tbreuer@ukaachen.de (T.B.); lmartin@ukaachen.de (L.M.); lara.stiehler@rwth-aachen.de (L.S.); gmarx@ukaachen.de (G.M.); tsimon@ukaachen.de (T.-P.S.); 3Institute of Molecular Cardiovascular Research, Medical Faculty Rheinisch-Westfälische Technische Hochschule Aachen University Hospital, D-52074 Aachen, Germany; 4Chair of Vascular Biology, Institute for Stroke and Dementia Research (ISD), Klinikum, Ludwig-Maximilians-Universität (LMU) München, 81377 Munich, Germany; Adrian.Hoffmann@med.uni-muenchen.de (A.H.); Juergen.Bernhagen@med.uni-muenchen.de (J.B.); 5Department of Anesthesiology, Klinikum der Universität München, Ludwig-Maximilians-Universität (LMU) München, 81377 Munich, Germany; 6Munich Cluster for Systems Neurology (SyNergy), 81377 Munich, Germany; 7Munich Heart Alliance, 80802 Munich, Germany; 8Department of Pneumology, Medical Faculty Rheinisch-Westfälische Technische Hochschule Aachen University Hospital, D-52074 Aachen, Germany; mdreher@ukaache.de; 9Department of Anaesthesiology, Intensive Care, Emergency and Pain Medicine, University Hospital, Wuerzburg, D-97080 Würzburg, Germany; cstoppe@ukaachen.de

**Keywords:** Macrophage Migration Inhibitory Factor (MIF), COVID-19, ICU treatment, acute respiratory distress syndrome (ARDS), SOFA Score, Horowitz Quotient

## Abstract

Mortality in critically ill coronavirus disease 2019 (COVID-19) patients is high and pharmacological treatment strategies remain limited. Early-stage predictive biomarkers are needed to identify patients with a high risk of severe clinical courses and to stratify treatment strategies. Macrophage migration inhibitory factor (MIF) was previously described as a potential predictor for the outcome of critically ill patients and for acute respiratory distress syndrome (ARDS), a hallmark of severe COVID-19 disease. This prospective observational study evaluates the predictive potential of MIF for the clinical outcome after severe COVID-19 infection. Plasma MIF concentrations were measured in 36 mechanically ventilated COVID-19 patients over three days after intensive care unit (ICU) admission. Increased compared to decreased MIF was significantly associated with aggravated organ function and a significantly lower 28-day survival (sequential organ failure assessment (SOFA) score; 8.2 ± 4.5 to 14.3 ± 3, *p* = 0.009 vs. 8.9 ± 1.9 to 12 ± 2, *p* = 0.296; survival: 56% vs. 93%; *p* = 0.003). Arterial hypertension was the predominant comorbidity in 85% of patients with increasing MIF concentrations (vs. decreasing MIF: 39%; *p* = 0.015). Without reaching significance, more patients with decreasing MIF were able to improve their ARDS status (*p* = 0.142). The identified association between an early MIF response, aggravation of organ function and 28-day survival may open future perspectives for biomarker-based diagnostic approaches for ICU management of COVID-19 patients.

## 1. Introduction

Until now, the coronavirus disease 2019 (COVID-19) pandemic has caused more than 2.1 million death worldwide and more than 100 million cases in total (update 28 January 2021) [1]. In Germany, mortality rates are high and the daily growing number of around 20,000–30,000 new cases during fall 2020 brought even intensive care unit (ICU) with large capacities to their limit. Reports about asymptomatic patients showed a wide range from 4–32%, due to the fact that it often remains unclear if these reports represent truly asymptomatic infection with never developing symptoms, or reports about patients who were free of symptoms by transmission to a hospital or medial service, but develop symptoms subsequently. However approximately 5% of patients with coronavirus 2 (SARS-CoV-2) infections, and 20% of those hospitalized, experience severe symptoms necessitating intensive care treatment [2]. The most common symptoms in hospitalized patients are fever (70–90%), dry cough (60–86%), shortness of breath (53–80%) [3], and ultimately the development of endothelial dysfunction and organ failure, that ultimately requires prolonged ICU treatment [4]. In particular the need for ICU treatment in cases of severe disease is related to high in-hospital mortality rates [5,6].

Acute respiratory distress syndrome (ARDS) is the most prevalent complication in hospitalized COVID-19 patients [7]. ARDS is characterized by an excessive influx of inflammatory cells, causing severe injury to alveolar endothelial and epithelial barriers. Consequently, an increased alveolar endothelial permeability results in leakage of a fluid, rich in proteins and inflammatory molecules, into the alveolar airspaces [8,9], resulting ultimately in the clinical presentation of ARDS.

Evidence from experimental and clinical studies suggests a pivotal role of macrophage migration inhibitory factor (MIF) in mediating and aggravating the pathogenesis of ARDS [10,11,12]. Its expression in cells and tissues involved in the front line of the host defense and stress response, its semi-constitutive expression and its rapid release from preformed intracellular pools as well as its glucocorticoid-overriding activity and cytokine-inducing capacity, position MIF as a central key player in controlling the immune and inflammatory response [13], which may be of particular relevance during the development of acute lung injury. MIF has been detected in alveolar macrophages in the lung tissue and alveolar airspace of ARDS patients [10]. In alveolar cells isolated from ARDS, MIF stimulated the release of the pro-inflammatory cytokine/chemokine interleukin-8 (IL-8/CXCL8) and the cytokine tumor necrosis factor-α (TNF-α), and additionally downregulated the immunosuppressive actions of glucocorticoids [10]. In ARDS animal models, anti-MIF-therapy proved to be beneficial and identified MIF as a potential therapeutic target [12,14]. Increased circulating MIF has been shown to be a potential biomarker to predict mortality in critically ill patients and patients with sepsis [15], whereas other studies illuminated its potential protective role during ischemia/reperfusion injury [16].

Accumulating evidence suggests that an increased mortality in COVID-19 patients correlates with a hyper-inflammatory and dysregulated response induced by ARDS in these patients [17], and alterations in the leukocytes and platelets count were described [18]. Increased serum levels of IL-6, IL-8 and TNF-α at the time of hospitalization have been found to be highly predictive for patient mortality [19,20]. Given the fact that fatal COVID-19 infections might be associated with an uncontrolled cytokine storm and given MIF’s upstream role in innate immunity and inflammation, we hypothesized that increased MIF levels might be associated with poor outcome and survival of COVID-19 patients.

Therefore, we analyzed the plasma concentration of MIF in a cohort study of COVID-19 patients during the ICU stay to evaluate whether circulating MIF plasma concentrations are related to clinically meaningful outcomes during ICU treatment. We surmised that the initiation of ICU treatment may result in changing plasma concentrations of MIF as a response to the life-supporting measures, and that these different reactions influence the outcome after ICU treatment.

## 2. Materials and Methods

### 2.1. Study Population and Data Collection

This prospective observational study was performed, based on ethical approval (EK 007/17, proofed on the 23 February 2017, and EK 100/20, proofed on the 7 April 2020), between the 16 March until the 13 April 2020 at the University Hospital of the Rheinisch-Westfälische Technische Hochschule Aachen, Germany. Legal representatives of all included patients provided written informed consent. Patients with positive SARS-CoV-2 test result, approved by real-time reverse transcription PCR (RT-PCR) from a nasal/pharyngeal swap, and ICU admission were consecutively enrolled in this study. Patients younger than 18 years, pregnant, or palliative care patients were excluded from the study. The standard care of the institutional standard procedures for treatment of COVID-19 patients was applied to the patients on ICU, including mechanical ventilation, initiation and use of extracorporeal membrane oxygenation (ECMO), and (if needed) renal replacement therapy (RRT). The necessity for ECMO therapy was decided for each patient individually on the basis of recently published Extracorporeal Life Support Organization (ELSO) consensus guideline [21]. Demographics, vital signs, laboratory values, blood gas analyses of each patient, as well as data for the organ support (ECMO and RRT) were extracted from the patient data management system (Intellispace Critical Care and Anesthesia (ICCA) system, Philips, Eindhoven, Netherlands). We evaluated the severity of the ARDS represented by the Horowitz quotient (HQ), the changes in global organ function, represented by the sequential organ failure assessment (SOFA) score, the need for ECMO, or RRT, respectively, dialysis over 14 days, as well as preexisting conditions of the included COVID-19 patients. Finally, the survival of all patients was monitored during a total ICU stay of 28 days.

### 2.2. Data Collection

We measured the plasma MIF concentration within the initial three days of ICU treatment and monitored the main outcome parameters during the following days, 28 days in total. Patients showing decreasing MIF values from day 1 to day 3 of ICU stay, were assigned to a group termed “Responder” to ICU treatment, whereas patients showing increasing MIF values from day 1 to day 3 were assigned to a group termed “Non-Responder” to ICU treatment.

All clinical and laboratory outcome parameters were collected from a chart review and performed as part of clinical routine. Laboratory evaluations were performed as indicated by the treating medical team and in accordance to the local institutional standards. At baseline, medical records and laboratory analyses were used to assess comorbidities and co-medication. Patients’ clinical and laboratory status was evaluated daily as indicated by the medical team and part of clinical standard. Evaluation of organ dysfunction (hemodynamic support, mechanical ventilation, renal replacement therapy) and possible adverse events was performed daily using clinical data and laboratory analyses to evaluate inflammation and organ dysfunction as per hospital routine. The main outcome parameters, which were evaluated for the Responder and Non-Responder groups included: 28-day survival, development of SOFA score and development of the severity of ARDS rated by the HQ in mild, moderate and severe ARDS. HQ is calculated as the ratio between partial oxygen pressure (paO_2_) and the inspiratory oxygen concentration (FiO_2_), displayed in mmHg. Additionally, the need for ECMO or RRT was assessed. Furthermore, blood gas analysis (BGA), cell counts, and ventilation parameters were observed. The general inflammatory status was monitored via C-reactive protein (CRP), procalcitonin (PCT) and IL-6 plasma concentrations, analyzed in an automated process of the in-house central laboratory as part of the clinical routine during the postoperative ICU treatment.

### 2.3. Measurement of Macrophage Migration Inhibitory Factor (MIF)

Citrated blood samples were withdrawn from the patients at enrollment to ICU (day1, d1), and on the third day of ICU treatment (day3, d3), centrifuged at 2000 g for 10 min, and the blood plasma was stored at −80 °C for subsequent analysis, using a commercially available MIF ELISA Kit (DMF00B, Human MIF ELISA Kit; R&D Systems Inc., Wiesbaden-Nordenstadt, Germany: Sensitivity: 0.068 ng/mL; Assay Range: 0.2–10 ng/mL).

### 2.4. Statistical Analysis

All statistical analyses were performed using the GraphPad Prism Software (GraphPad Software 9.0, La Jolla, CA, USA). All data was checked for normal distribution using a Shapiro-Wilk test. Differences in MIF concentrations between day 1 and day 3 were calculated using a paired *t*-test. Changes in MIF concentrations between day 1 and day 3 were compared by one-way analysis of variance (ANOVA) for repeated measures with a Tukey post hoc analysis for multiple comparison. Distribution of survival, gender and age distribution, as well as SOFA score improvement, ARDS severity according to the Horowitz Quotient, and prevalence of ECMO or RRT was checked for significant differences between the patient groups, using a two-sided Fisher’s exact test. Data are displayed as mean ± standard deviation (SD). All statistical tests were 2-tailed and a two-sided *p*-value of 0.05 was considered for significance.

## 3. Results

### 3.1. Included Patients

All mechanical ventilated COVID-19 patients, enrolled to ICU between 16 March and 13 April 2020, with a complete data set for MIF analysis (day1 and day3), were included and observed for 28 days. This collective was represented by 26 male and 10 female patients (36 in total), at the age of 62 ± 8 years, by their admission to ICU (day 1). Patients with partial (*n* = 11) ventilation or without need for ventilation (*n* = 7) within this time frame, or with incomplete data sets (day1 and day3) for MIF analysis, were excluded from this study. Overall, circulating plasma MIF concentrations were determined from 36 COVD-19 patients on the 1st and the 3rd day during the ICU stay.

### 3.2. Plasma MIF Concentration and Grouping of Patients

The overall time course of circulating MIF levels did not show significant differences between day 1 (admission day) and day 3 after ICU admission (Figure 1).

When considering the individual time courses of MIF concentrations measured for each patient, between day 1 and day 3, we revealed two different groups, with opposing development of MIF concentration. Among these, a patient group (*n* = 18) with significantly decreasing values from day 1 (11.6 ± 8.2 ng/mL) to day 3 (5.4 ± 3.3 ng/mL, *p* < 0.001) could be identified (Responder), whereas the other patient group (*n* = 18) showed a significant increase of MIF (Non-Responder) values from day 1 (5.9 ± 3.7 ng/mL) to day 3 (9.8 ± 4.62 ng/mL, *p* < 0.001). The concentration on day 1 and day 3 was significantly different (5.4 ± 3.3 vs. 9.8 ± 4.62 ng/mL, *p* < 0.001) between these patient groups (Figure 2).

### 3.3. Preexisting Conditions

All included patients were checked for preexisting conditions and for differences in the frequency of these conditions between the two groups (Responder vs. Non- Responder). Except for hypertension as preexisting condition, no relevant differences could be reported between the patient groups (Table 1).

### 3.4. The Impact of Circulating MIF Levels on Severity of Organ Injury Measured by Sequential Organ Failure Assessment (SOFA) Score

Reduced SOFA score at day 14 in relation to day 1 indicates a recovery of the patients, respectively an improved global organ function, whereas an increasing SOFA score at day 14 in comparison to day 1 indicates an aggravation of the patient organ function. No differences concerning the number of patients showing an improved-, aggravated- or equal-organ function after 14 days could be observed between the groups. The SOFA score on day 14 was significantly higher in the patients showing aggravated organ function, compared to the patients with improved organ function, without differences between the groups (Responder day 14: 7 ± 2.4 vs. 12.6 ± 2; *p* < 0.01/Non-Responder day 14: 8.1 ± 2 vs. 14.3 ± 3; *p* < 0.01). However, within the Non-Responder group, the SOFA score in those patients showing aggravated organ function, increased significantly from day 1 to day 14 whereas the increase in the Responder group was not significant (8.2 ± 4.5 to 14.3 ± 3; *p* < 0.01) (Figure 3). No correlation between an aggravation of organ function and 28 days’ survival could be shown (*p* = 0.167).

### 3.5. The Influence of Circulating MIF Levels on Oxygenation Rate as Measured by Horowitz Quotient

The number of patients changing their ARDS status from day 1 to day 14 was not different between the groups (*p* = 0.142). Also, within the groups, no significant changes concerning the ARDS status could be detected. However, a clear tendency towards improved HQ, and transition of patients from severe to moderate or mild ARDS could be detected during the observation period in the Responder group (*p* = 0.146), compared to the Non-Responder group (*p* = 0.370). Although fewer patients in the Non-Responder group showed a transition from severe to mild or moderate or mild ARDS, no correlation between ARDS severity and the 28-day survival could be shown for any of the groups (*p* = 0.146) (Figure 4).

### 3.6. The Association between MIF Levels and Extracorporeal Membrane Oxygenation (ECMO) and Renal Replacement Therapy 

Regarding the use of ECMO or renal replacement therapy, no difference could be detected between the Responder or Non-Responder groups (ECMO Responder vs. Non-Responder *p* = 0.248 / RRT Responder vs. Non-Responder *p* = 0.494).

### 3.7. The Association between MIF Levels and Blood Gases, Cell Count and Ventilation Parameters

No differences between the groups, at any time during the observational period could be detected for the blood gas analysis (pH: *p* = 0.841; pCO_2_: *p* = 0.929; pO_2_: *p* = 0.924; lactate: *p* = 0.728; SpO_2_: *p* = 0.927), the blood cell counts (red blood cells: *p* = 0.875; white blood cells: *p* = 0.888; platelets: *p* = 0.511), or the ventilation parameters (tidal volume: *p* = 0.998; positive end expiratory pressure: *p* = 0.846; respiratory frequency: *p* = 0.764).

### 3.8. Pro-Inflammatory Markers

No relevant differences between the patients‘ groups were detected regarding the plasma levels of other pro-inflammatory markers and cytokines such as interleukin 6 (IL-6), procalcitonin (PCT) or C-reactive protein (CRP) during 14 days of ICU treatment (Figure 5).

### 3.9. The Association between MIF Values and 28-Day Survival on Intensive Care Unit (ICU)

Comparing the patients‘ group of “Responder” and “Non-Responder”, the survival after 28 days showed a significant difference between these patient groups; 93% of the Responder group survived 28 days ICU treatment, whereas only 56% of the Non-Responder patients survived (*p* < 0.01) (Figure 6A). Additionally, the MIF concentration on day 3 within the Non-Responder group was significantly increased in patients who died, when compared to the survivors (12.6 ± 3.7 ng/mL vs. 7.6 ± 4.5 ng/mL; *p* = 0.026). In the Responder group, only one patient died, nevertheless, also this MIF concentration was higher (Figure 6B). No difference in gender distribution per se could be found between the groups. However, significantly more female patients died during the 28 days’ observational period on ICU in the Non-Responder group, compared to the Responder group (Figure 6C). No differences could be found in the average age of the patients between the groups.

## 4. Discussion

The currently ongoing COVID-19 pandemic challenges health care systems and ICUs worldwide. There remains an urgent need to identify early predictive biomarkers to identify patients that have a high risk for severe disease progression and mortality especially in the light of highly variable courses of disease and limited ICU capacities [2,22]. Pharmacological treatment strategies next to ICU supportive care are rare so that a biomarker-based early stage reevaluation of ICU therapy could improve patient outcomes by facilitating treatment choices [23]. Severe progression of disease seems to be triggered by an overactivated state of the adaptive immune system and a cytokine release syndrome with IL-6 and IL-10 as potential predictive cytokine markers [22,24]. MIF is one of the inflammatory cytokines that contributes to the “cytokine storm” and has been shown to be a potential biomarker and therapeutic target in ARDS [10,11,12].

Focusing on the early phase of ICU treatment, we measured plasma MIF concentrations in COVID-19 patients at day 1 and 3 after ICU enrollment. We evaluated patient outcome by assessing SOFA Score, HQ, ECMO, RRT and inflammation markers at day 14 as well as survival 28 days after admission. The detected plasma levels of MIF reveal two groups of patients; one group with increasing concentrations, and one group showing decreasing plasma concentrations of MIF between the first and third observational day. We defined decreasing concentrations as a positive reaction to ICU treatment and summarized these patients in a collective as “Responder” group, whereas the increasing concentrations were defined as no reaction, respectively a “Non-Responder” group.

We observed a strong correlation between the Non-Responder group with increasing MIF levels and mortality and, respectively, the Responder group with declining plasma MIF concentrations and survival. Additionally, we could show that the aggravation of organ function and, respectively the increase of SOFA score in patients of the Non-Responder group, was worse and significantly stronger, whereas the aggravation of organ function within the Responder group was milder and not significant. Furthermore, a strong tendency towards an improved HQ, representing an improved ARDS category, could be shown for the Responder group, whereas fewer patients in the Non-Responder group showed a tendency towards improved ARDS category within 14 days of ICU treatment. Finally, hypertension could be identified as one preexisting condition, significantly related to the patients in the Non-Responder group

As reported previously, circulating MIF concentrations, as well as circulating MIF-2 concentrations, have the potential to predict the outcome of critically ill patients as a biomarker [15]. D-dopachrome tautomerase is defined as MIF-2 or DTT, and is a member of the MIF superfamily [25]. Both markers, MIF and MIF-2, have been found to be increased in ICU patients in comparison to healthy controls and in 30 septic patients in comparison to 42 non-septic patients. High concentrations of both markers were positively correlated with non-survival; respectively, low concentrations were related to survival [15]. Our data strongly support this finding in septic patients, now also for critically ill COVID-19 patients, suffering from ARDS, as we could measure significantly higher MIF concentrations in the non-survivors when compared to the survivors.

Recent studies, using a combination of inflammatory biomarkers, identified two main sub-phenotypes of ARDS, namely the hyper-inflammatory phenotype and the low-inflammatory phenotype, significantly differing in survival and organ function [22,26]. In line with our findings, Bime et al. identified MIF as one out of six biomarkers (angiopoietin-2, MIF, IL-8, IL-1 receptor antagonist, IL-6, and extracellular nicotinamide phosphoribosyltransferase) and, when combined, as a strong predictor of 28-day mortality in ARDS patients [27].

However, our data did not include the whole data set of the biomarker-based predicted model of Bime et al. except for IL- 6 and MIF. In contrast to the ARDS studies, using inflammatory biomarkers for sub-classification of ARDS [27], as well as the most recent COVID-19 studies [28,29], demonstrating that IL-6 correlates with severe and fatal ARDS/COVID-19, we found no differences in serum IL-6 concentrations between Responder and Non-Responder groups. Furthermore, the concentration of the acute phase proteins PCT and CRP were not significantly different between Responder and Non-Responder groups at any time during the 14 days. The missing effects of these inflammatory markers might be related to incomplete datasets and very high standard deviations in the measured concentrations of IL-6, CRP and PCT. In comparison to the large deviation within the measurements of IL-6, PCT and CRP, the data for the MIF concentration were much more consistent, showed clear differences between the groups, and appear as a reliable dataset. Nevertheless, the assessment of additional biomarkers, might be a useful tool to classify patients into hyper- and low-inflammatory phenotypes as well as determination of survival probability. However, our MIF data suggest that the COVID-19 patients with a decline of serum MIF within 3 days after ICU admission (Responder) might belong to the low-inflammatory phenotype, while critically ill patients with an MIF increase (Non-Responder) might be assigned to a hyper-inflammatory phenotype, which is associated with a high mortality. This hypothesis is further strengthened by the HQ we evaluated over 14 days. Although the number of patients showing an improvement of the ARDS category within the Responder group did not differ significantly from the number of patients in the Non-Responder group within this observational period, a strong tendency towards recovery could be shown in the Responder group (low-inflammatory phenotype), which is missing in comparison to the Non-Responder group (high-inflammatory phenotype). However, the ARDS severity did not correlate with the significantly higher mortality rate in the Non-Responder group, which is in contrast to a recent multi-center, prospective cohort study which reported that mortality was significantly higher in COVID-19 patients presenting with severe ARDS [30]. This discrepancy to our findings might be due to the small number of patients included in our study. Additionally, pulmonary MIF levels such as obtained from bronchoalveolar lavage or tissue specimens reflecting local MIF concentrations were not investigated in this study.

Assuming that our patients are, rather, classified into the recently identified low- and high-inflammatory sub-phenotype of ARDS instead of ARDS severity (Berlin definition), the lack of significant differences in HQ categorization between the Responder and Non-Responder groups, as well as missing relevant differences in organ dysfunction (SOFA score) between the groups are in accordance with the findings of Bime et al. They demonstrated that these phenotypes cannot be distinguished by using traditional parameters such as Horowitz score (severity of ARDS (PaO_2_/FiO_2_ ratio), severity of renal or hepatic failure, or the extent of leukocytosis. Regardless of the missing significant differences between the groups, within the Non-Responder group, the intensity of SOFA score increase was significantly higher, whereas the SOFA score increase within the Responder group did not reach significance. These findings support the categorization of our Non-Responder group into a high-inflammatory phenotype with worse organ function and survival outcome for the patients.

Only one pre-existing condition, namely hypertension, was strongly associated with the Non-Responder, respectively the high-level MIF group and higher mortality for ARDS patients with SARS-CoV-2 infection. A recent meta-analysis confirmed hypertension as relevant comorbidity for severe COVID-19 infections in comparison to non-severe (odds ratio (OR) 2.36) [31], which is in a row with our findings, although we could not confirm further comorbidities which were reported as relevant in the meta-analysis of Yang et. al., like diabetes, and cardiovascular diseases, for example. However, hypertension could not be proven to be an independent risk factor for the development of a severe progression of COVID-19 yet [32], and also MIF was not described as an independent biomarker for ARDS and, respectively, severe COVID-19 infection.

Summarizing, and although we only conducted a relatively small cohort observational study, we demonstrated that COVID-19 patients with an increase of MIF levels (high inflammatory phenotypes) within three days after admission exhibited a higher mortality than those with a decline in MIF concentration (low-inflammatory phenotypes). Additionally, a stronger increase of SOFA score and, respectively worse organ function, could be shown for the patients with increasing MIF concentrations, and a clear tendency towards ARDS recovery was only detectable in patients with decreasing MIF concentrations. Furthermore, hypertension as comorbidity and increasing MIF concentration as an early biomarker could be related to increased mortality in COVID-19 patients in our study.

The study has several limitations which need to be considered carefully when interpreting these data. First, data were received from a small cohort of patients, showing high deviations in some of the measurements, and due to lacking data sets for not, or partial, mechanically ventilated COVID-19 patients, we were not able to include these groups in our analysis. However, given the clinical relevance of COVID disease, even explorative data such as these are urgently needed to investigate new diagnostic and treatment strategies. Second, the received results obtained from our observational, clinical study remain correlative and do not necessarily show a causative relationship. Follow-up large-scale clinical studies to validate the clinical significance of MIF serum levels will be required. 

## 5. Conclusions

The present findings highlight a potential role for MIF as an early biomarker, which may identify a successful response to the initiated ICU treatment bundles in critically ill patients with COVID-19 disease. The identified association between the MIF response, organ function and survival may open future perspectives for diagnostic approaches or to monitor the success of initiated treatment strategies.

## Figures and Tables

**Figure 1 diagnostics-11-00332-f001:**
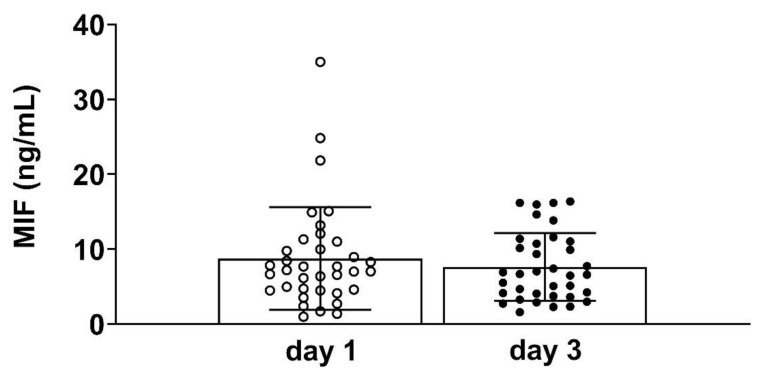
Plasma concentrations of macrophage migration inhibitory factor (MIF) on day 1 and day 3 of Intensive Care Unit (ICU) admission in 36 patients (day 1 vs. day 3, *p* = 0.307).

**Figure 2 diagnostics-11-00332-f002:**
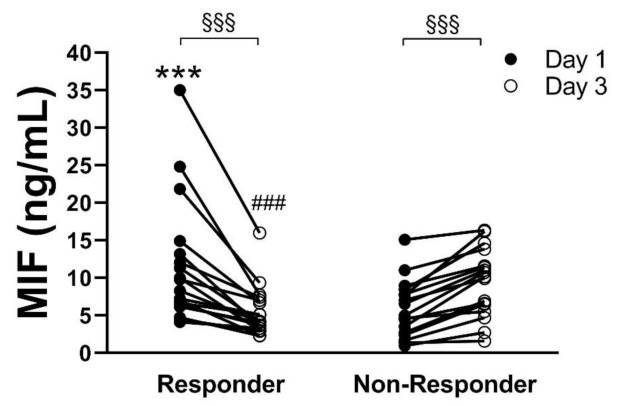
Plasma concentration of macrophage migration inhibitory factor (MIF) on day 1 (black circles) and day 3 (white circles) of intensive care unit (ICU) admission for each of the 36 coronavirus disease 2019 (COVID-19) patients. Patients showing decreasing MIF concentrations between day 1 and day 3 were summarized as “Responder” and those with increasing MIF concentrations were summarized as “Non-Responder” *** *p* < 0.001 vs. day 1 Non-Responder; ^###^
*p* < 0.001 vs. day 3 Non-Responder; ^§§§^
*p* < 0.001 day 1 vs. day 3.

**Figure 3 diagnostics-11-00332-f003:**
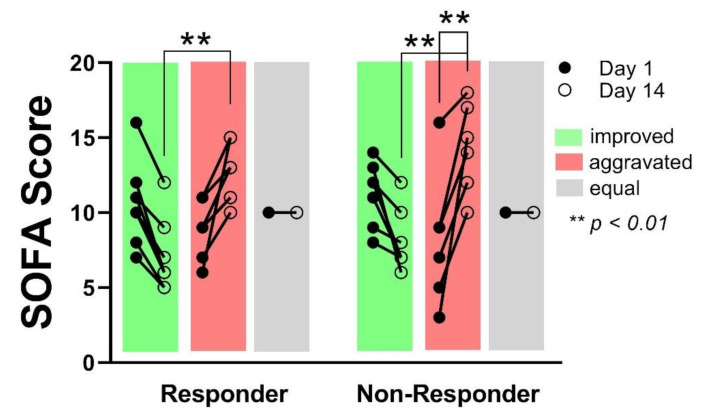
In both groups, an equal number of patients showed an improved sequential organ failure assessment sequential organ failure assessment (SOFA) score, respectively improved organ function after 14 days (green), as well as an aggravated (red), or unchanged (grey) organ function. SOFA score on day 14 was significantly different between improved- and aggravated organ function, independent from the groups (Responder vs. Non-Responder). Within the Non- Responder group, the SOFA score increase was significant from day 1 to day 14 for patients with aggravated organ function.

**Figure 4 diagnostics-11-00332-f004:**
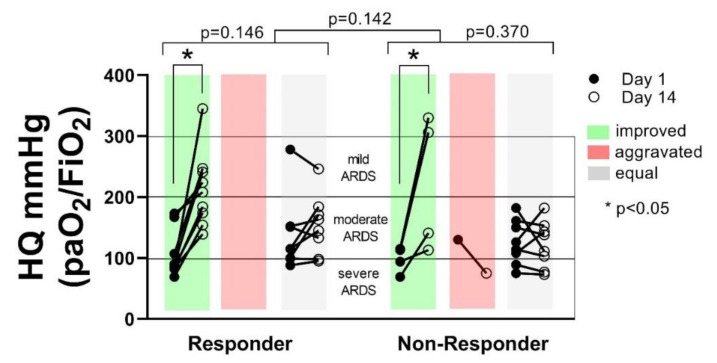
The number of patients changing their acute respiratory distress syndrome (ARDS) status from improved (green), aggravated (red) or equal (grey) from day 1 to day 14 was not significantly different between the groups (Responder vs. Non-Responder *p* = 0.142). Within the groups, the number of patients changing their ARDS status was not significantly different (*p* = 0.146 vs. *p* = 0.370). In both groups, a significant increase of HQ in patients showing improvement from day 1 to day 14 could be shown (*p* < 0.05).

**Figure 5 diagnostics-11-00332-f005:**
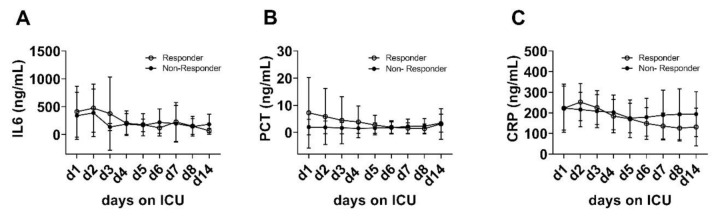
No changes of plasma interleukin 6 (IL-6) (**A**), procalcitonin (PCT) (**B**) and C-reactive protein (CRP) (**C**) from day 1 to day 14 were detected within or between the groups (IL-6: *p* = 0.587; PCT: *p* = 0.228; CRP: *p* = 0.955).

**Figure 6 diagnostics-11-00332-f006:**
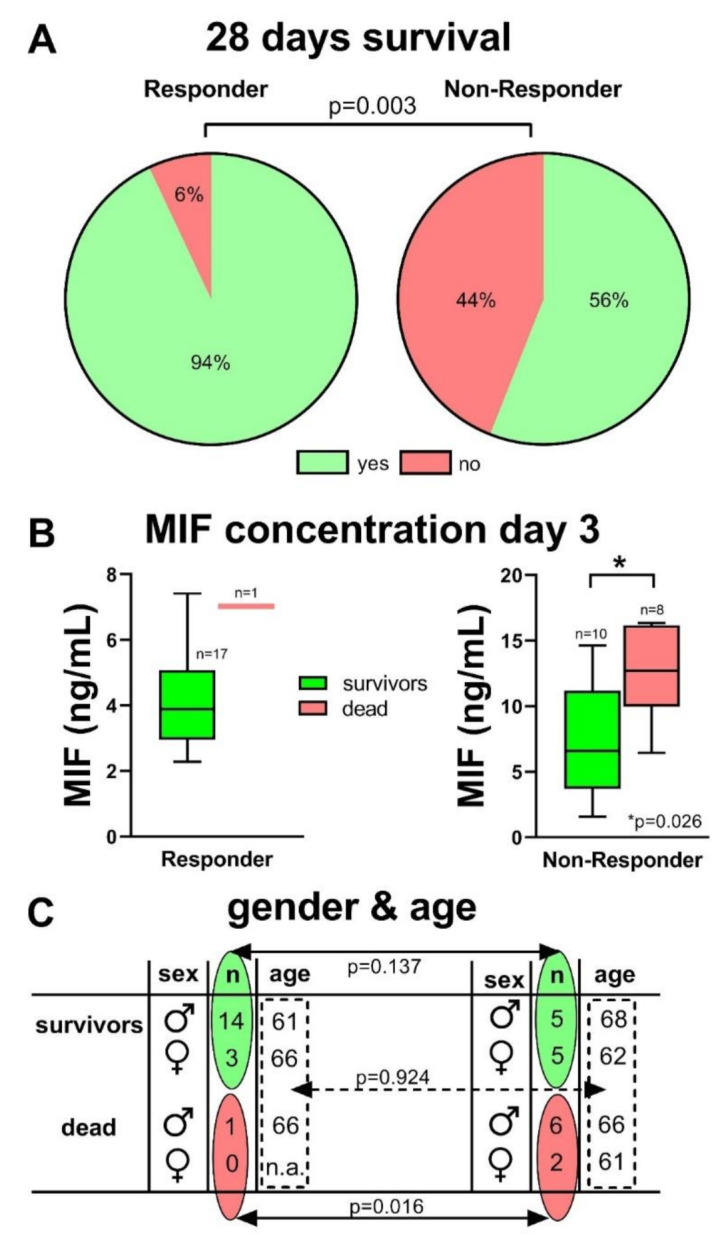
Significantly more COVID-19 patients survived the 28 days ICU treatment in the Responder group (**A**, *p* = 0.003). Within the Non-Responder group, the MIF concentration on day 3 was significantly increased in patients who died compared to survivors (*p* = 0.026) (**B**). There was no significant difference in gender- (*p* = 0.137) and age-distribution (*p* = 0.924), but none of the female patients (*n* = 3) died in the Responder group, whereas 2 of 7 female patients in the Non-Responder group died during the observational period (*p* = 0.0159) (**C**).

**Table 1 diagnostics-11-00332-t001:** Significantly more coronavirus disease 2019 (COVID-19) patients in the Non-Responder group suffered from hypertension by the time of ICU enrollment (* *p* = 0.0153). The remaining preexisting conditions are equally distributed between the groups.

Preexisting Condition/Comorbidities	Responder *n* = 18 Comorbidities (%)	Non-Responder *n* = 18 Comorbidities (%)	*p*-Value
Hypertension	*n* = 7 (39)	*n* = 15 (84)	0.0153 *
Diabetes	*n* = 5 (28)	*n* = 6 (33)	0.999
Obesity	*n* = 3 (17)	*n* = 7 (39)	0.264
Ischemic heart disease	*n* = 3 (17)	*n* = 5 (33)	0.691
Chronic obstructive pulmonary disease	*n* = 1 (6)	*n* = 2 (11)	0.999
Tumor	*n* = 1 (6)	*n* = 1 (6)	0.999
Chronic kidney disease	*n* = 4 (22)	*n* = 4 (22)	0.999
Thromboembolic events	*n* = 2 (11)	*n* = 1 (6)	0.999
Nicotine abusus	*n* = 1 (6)	*n* = 1 (6)	0.999
Hepatitis B	*n* = 0 (0)	*n* = 2 (11)	0.486

## Data Availability

The data presented in this study are available on request from the corresponding author. The data are not publicly available due to personalized data of patients included to this study and due to the medical confidentiality.

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
