# Peer review of "Macrophage Migration Inhibitory Factor (MIF) Plasma Concentration in Critically Ill COVID-19 Patients: A Prospective Observational Study"

_diagnostics, 2021, doi:10.3390/diagnostics11020332_

Round 1

Reviewer 1 Report

A very well-written manuscript in which the authors attempted to assess the role of MIF in patients with COVID-19 in the intensive care unit.

The article does not raise any methodological or linguistic issues.

A few minor notes below:
1. ELISA kit - please provide CVs. detection limit or other data, if provided by the manufacturer.
2. in the methodology, please add information on how the concentration of other parameters (IL-6, PCT, CRP) was assessed.
3.below I enclose a proposed literature that may be added by the authors:

Res Sq [Preprint]. 2021 Jan 14:rs.3.rs-141578. doi: 10.21203/rs.3.rs-141578/v1. PMID: 33469573; PMCID: PMC7814832.

Pathogens. 2020 Jun 20;9(6):493. doi: 10.3390/pathogens9060493. PMID: 32575786; PMCID: PMC7350358.

Cytokine. 2021 Jan 15;140:155438. doi: 10.1016/j.cyto.2021.155438. Epub ahead of print. PMID: 33493861

Author Response

The authors thank the Reviewer for the very kind commentary on the manuscript, and the important notes. We answered the notes as follows, and added the parts in red color to the revised manuscript file, which you can find attached as pdf file:

  1. ELISA kit - please provide CVs. detection limit or other data, if provided by the manufacturer.

Thank you for this note, the authors added the following information on page 4, line 148+149: Sensitivity: 0.068ng/mL; Assay Range: 0.2 – 10ng/mL.

  1. in the methodology, please add information on how the concentration of other parameters (IL-6, PCT, CRP) was assessed.

Thank you for this important note, the authors added the following information on page 4, line 141:

Furthermore, Blood Gas Analysis (BGA), cell counts, and ventilation parameters were observed. The general inflammatory status was monitored via C-reactive protein (CRP), Procalcitonin (PCT) and IL-6 plasma concentrations, analyzed in an automized process of the in-house central laboratory as part of the clinical routine during the postoperative ICU treatment.

  1. below I enclose a proposed literature that may be added by the authors:

The authors thank the reviewer for this valuable update, and we added these relevant references as follows:

Page 2, line 85:

Reference 17: Res Sq [Preprint]. 2021 Jan 14:rs.3.rs-141578. doi: 10.21203/rs.3.rs-141578/v1. PMID: 33469573; PMCID: PMC7814832.

Page 2, line 86:

Reference 18: Pathogens. 2020 Jun 20;9(6):493. doi: 10.3390/pathogens9060493. PMID: 32575786; PMCID: PMC7350358.

Page 2, line 88:

Reference 20: Cytokine. 2021 Jan 15;140:155438. doi: 10.1016/j.cyto.2021.155438. Epub ahead of print. PMID: 33493861

Reviewer 2 Report

Authors of the manuscript raise an important and interesting issue.

In my opinion the numer of COVID-19 deaths should be updated (page 1, lines 46 - 47).

The study is carefully conducted and the statistical method is right, however, the number of cases is small. The manuscript is written in a good style and easy to read, there are no grammatical,  punctuation or linguistic errors. The discussion is written correctly, addresses all the most important issues as well as practical implications of the conducted research.

Author Response

The authors thank the reviewer for the very kind commentary on the manuscript. We answered the important note as follows and added the parts in red color to the revised manuscript file, which you can find attached as pdf file:

  1. In my opinion the number of COVID-19 deaths should be updated (page 1, lines 46 - 47).

Thank you for this important note, we added the actual numbers from the cited COVID-19 Dashboard:

Page 2, line 51-52: Until today, the COVID-19 pandemic caused more than 2.1 million death worldwide and more than 100 million cases in total [1].